# IDENTIFYING NEURAL DYNAMICS USING INTERVENTIONAL STATE SPACE MODELS

## ABSTRACT

Neural circuits produce signals that are complex and nonlinear. To facilitate the understanding of neural dynamics, a popular approach is to fit state space models (SSM) to data and analyze the dynamics of the low-dimensional latent variables. Despite the power of SSM in explaining neural circuit dynamics, it has been shown that these models merely capture statistical associations in the data and cannot be causally interpreted. Therefore, an important research problem is to build models that can predict neural dynamics under causal manipulations. Here, we propose interventional state space models (iSSM), a class of causal models that can predict neural responses to novel perturbations. We draw on recent advances in causal dynamical systems and present theoretical results for the identifiability of iSSM. In simulations of the motor cortex, we show that iSSM can recover the true latents and the underlying dynamics. In addition, we illustrate two applications of iSSM in biological datasets. First, we apply iSSM to a dataset of calcium recordings from ALM neurons in mice during photostimulation and uncover dynamical mechanisms underlying short-term memory. Second, we apply iSSM to a dataset of electrophysiological recordings from macaque dlPFC recordings during micro-stimulation and show that it successfully predicts responses to unseen perturbations.

## 1 INTRODUCTION

Understanding neural data requires identifying dynamics underlying it. The principled way to achieve this is through causal perturbations. When a perturbation is delivered, the activity of perturbed neurons is dissociated from their upstream neurons, facilitating the inspection of the circuit dynamics when certain edges are functionally removed from the circuit. This powerful strategy enables testing sophisticated neural hypotheses. For example, O'Shea et al. (2022) uses perturbations to understand whether dynamics in the motor cortex are path-following (driven by an upstream brain region), low-dimensional, or high-dimensional. Another example by Feulner et al. (2022) uses a similar strategy to investigate whether feedback drives plasticity for rapid learning in the motor cortex. Another study by Sanzeni et al. (2023) uses optogenetic perturbations to uncover the degree of coupling in the visual cortex of mice and monkeys. They show through modeling that under strong network coupling, the perturbations lead to a reshuffling of responses in the circuit.

The main insight of these works is that in the absence of perturbations (i.e. *observational regime*), neural dynamics are confined to low-dimensional spaces, and models that are built upon observational data are not able to capture neural dynamics outside of the low-dimensional space. However, during perturbations (i.e. *interventional regime*), the neural state is driven outside of the task space providing more information about dynamics in the global neural state space (Jazayeri & Afraz, 2017). This insight allows us to build sophisticated hypotheses that can only be tested using perturbations (Fig. 1). Interventional studies are critical for determining the causal contribution of neural dynamics to behavior and perception. For example, a study by Shahbazi et al. (2022) uses electrical stimulation to manipulate a monkey's perception using targeted stimulation.

Here, we rest on these ideas and develop a new class of latent variable models that aim to capture neural dynamics in both observational and interventional regimes. We base our model on the framework of Causal Inference (CI) (Pearl et al., 2016). Instead of directly modeling the joint distribution of the data, CI uses structural equations to describe the generative process of the data. In

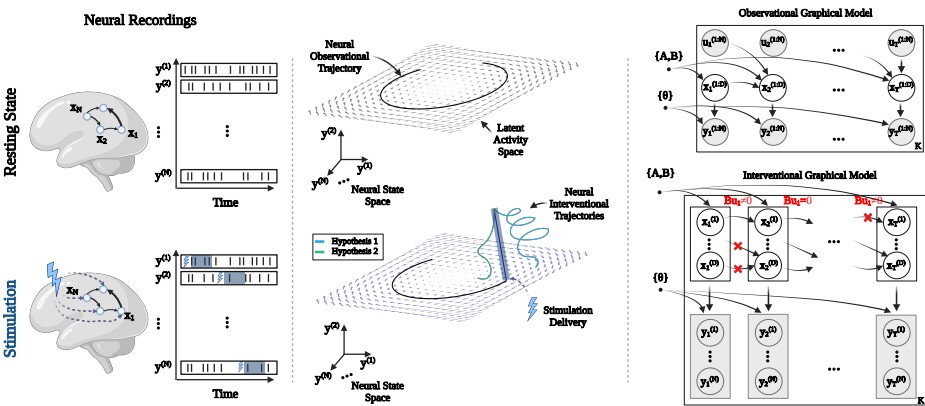

Figure 1: **Overview.** Neural dynamics in observational (top) and interventional (bottom) regimes. The observational data is confined to a low-dimensional task space whereas the interventional data explores the state space enabling the testing of causal neural hypotheses.

this framework, interventions are modeled as changing the structural equations. Equivalently, when an intervention is performed on a node, it is disconnected from all its parents in the generative model, and its distribution is set to a new distribution. A major benefit of modeling the interventions in this way is that having access to interventional data allows us to identify the model parameters as we will see in section 3.3.

Many of the popular models used in neuroscience suffer from identifiability issues (Maheswaranathan et al., 2019). In one line of work, researchers have developed similarity metrics that are agnostic to non-identifiability transformations (see Sucholutsky et al. (2023) for a review). However, in many cases, the model parameters or latent variables are biologically meaningful, and recovering them is desired. Therefore, the need for developing identifiable models for neuroscience data analysis is an overarching goal. For example, Zhou & Wei (2020) use an identifiable VAE as opposed to a vanilla VAE and infer latent variables that encode the geometry of the task in an unsupervised manner.

When modeling time series, specific challenges are involved. These challenges appear both at the level of structural equations and in modeling the interventions. We will describe our modeling framework in section 3.1.

## 2 RELATED WORK

**State Space Models**   To understand neural circuits, a popular strategy is to build low-dimensional state space models (SSM). Driven by the neural manifold hypothesis, neuroscientists often assume that neural data lies on a low-dimensional manifold. The challenge then becomes discovering the latent manifold and characterizing how the dynamics evolve in the low-dimensional space. Subsequently, a suite of SSMs have been developed covering a wide range of assumptions and applications. A typical SSM follows a dynamic model and an emission model described by the following equations:

$$\boldsymbol{x}_{t+1} = g_{\boldsymbol{\theta}}(\boldsymbol{x}_t) + \boldsymbol{\epsilon}_t, \quad \boldsymbol{y}_t = f_{\boldsymbol{\theta}}(\boldsymbol{x}_t) + \boldsymbol{\delta}_t, \quad \boldsymbol{\epsilon}_t \sim p(\boldsymbol{\epsilon}_t), \quad \boldsymbol{\delta}_t \sim p(\boldsymbol{\delta}_t).$$

With this general formulation, models depart based on the specification of $g_{\boldsymbol{\theta}}, f_{\boldsymbol{\theta}}, p(\boldsymbol{\epsilon}_t), p(\boldsymbol{\delta}_t)$. Linear dynamical systems (LDS) assume that both $g_{\boldsymbol{\theta}}, f_{\boldsymbol{\theta}}$ are linear and $p(\boldsymbol{\epsilon}_t), p(\boldsymbol{\delta}_t)$ are multivariate normal distributions. A separate line of work assumes that $g$ is switching linear and develops algorithms that jointly infer switching times as well as latent states (Petreska et al., 2011; Linderman et al., 2017; Fox et al., 2008). These models have been successful in particular when there are abrupt changes in the dynamics.

LDS is known to have a limited capacity to express complex datasets. A method known as P$f$LDS (Gao et al., 2016) extends the LDS model by replacing its linear emission model with an arbitrary nonlinear transformation followed by Poisson noise. It has been argued theoretically that a linear dynamical system (in a latent space with sufficiently large dimension) followed by a

nonlinear emission is powerful to model any dynamical system (Koopman, 1931). Therefore P$f$LDS has the capacity to fit complex datasets.

Although SSMs have been primarily used for fitting observational data, there has been a few attempts applying them to interventional data as well. However, responses to perturbations are often modeled as additive which makes the SSM models non-causal. We will elaborate on this further in Section 3.1. here we extend upon SSM and provide a complementary view from a causal perspective.

**Model Identification in Static Data** The emerging field of causal representation learning provides statistical treatments for recovering the true parameters of statistical models. Most of the developments correspond to static models and can be broadly categorized into identification using observational or interventional data. **(1) Observational:** While early theoretical guarantees have been limited to linear mixing and asymmetric noise (Comon, 1994), these results have been extended to nonlinear mixing (Locatello et al., 2019; Xi & Bloem-Reddy, 2023), and nonlinear mixing with observation noise (e.g. VAEs) (Khemakhem et al., 2020), and multi-environment data (Lachapelle et al., 2023). **(2) Interventional:** With access to interventional data, identifiability results can be extended to broader classes of models. Lippe et al. (2022) show that with sparse interventions we can recover latents up to permutation, scaling, and offset. Ahuja et al. (2023) utilize independent support properties (Wang & Jordan, 2021) and provide identifiability guarantees. These results have been further extended to nonparametric latents with linear and nonlinear mixing functions (von Kügelgen et al., 2023; Buchholz et al., 2023; Varici et al., 2023).

**Model Identification in Dynamic Data** More recently theoretical results on statistical model identification have been extended to Markov models and switching linear dynamical systems (Balsells-Rodas et al., 2023). These results provide the identification of the model parameters up to a class of nuisance transformations (e.g. affine). Most relevant to our work are Yao et al. (2022; 2021). The main shortcoming of these works is that they do not incorporate noise in the observation space, which is crucial for modeling biological datasets. Previous work can be broadly categorized into two groups. Some studies consider the transient interventional effects while others investigate the persistent effects in the stationary regime (Schölkopf & von Kügelgen, 2022; Malinsky & Spirtes, 2018; Besserve & Schölkopf, 2022; Benkő et al., 2018; Malinsky & Spirtes, 2018; Peters et al., 2022). Hansen & Sokol (2014) uses differential equations as structural equations in dynamical systems. Ahuja et al. (2021) considers (deterministic) linear dynamics (referred to as mechanism) and nonlinear emissions (referred to as rendering) and proves that the latent space of such a model is identifiable from observational data up to mechanism invariances. Lippe et al. (2023) show that for linear dynamics, if we have access to binary interventional data then the latents are identifiable up to permutation. Yao et al. (2022); Song et al. (2024) focus on the identification of latent non-stationary dynamics using observational data. Hyvarinen & Morioka (2016; 2017); Hyvarinen et al. (2019); Hälvä et al. (2021) focus on extending nonlinear ICA and its identifiability results to temporal settings. They impose constraints on the mixing function and the latent dynamics to achieve identification using only observational data.

In addition to the statistical literature on model identification, recent work in dynamical systems theory has utilized the Koopman theory to find conditions such as sampling frequency for the exact identification of the continuous time dynamical systems from sampled data (Zeng et al., 2022).

## 3 METHODS

### 3.1 INTERVENTIONAL STATE SPACE MODELS

Consider an experiment with $N$ recorded neurons over $T$ time steps repeated for $K$ trials. We denote neural responses at time $t$ by $\boldsymbol{y}_t$ where $\boldsymbol{y}_t$ is a $N$-vector that concatenates the spike counts or calcium activities of all neurons. We assume the existence of a time-dependent latent variable $\boldsymbol{x}_t \in \mathbb{R}^D$ where $D$ is the dimension of latent space. We present the interventional model and elaborate on its difference with the observational model.

The first modeling assumption that distinguishes iSSM from SSM is that we assume perturbing neurons affects the latent dynamics directly, which will consequently affect neural responses in the next time point according to the emissions model. The second more critical assumption is

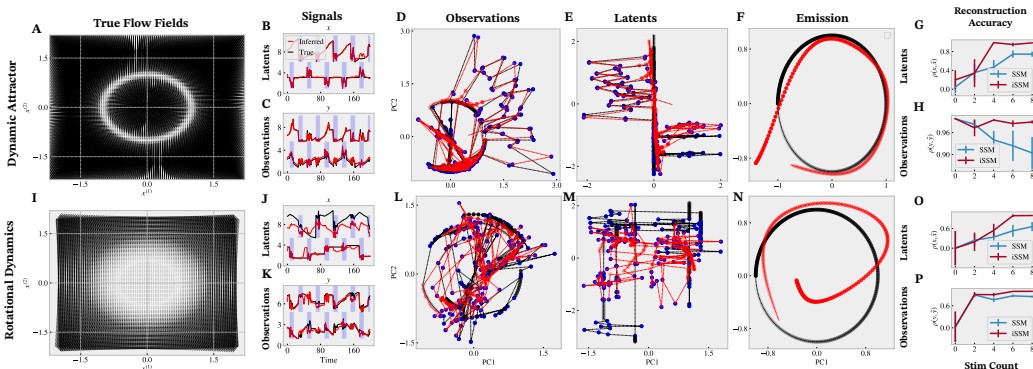

Figure 2: **Results on Models of Motor Dynamics.** (A) Flow field underlying dynamic attractor model of motor cortex. (B,C) True (black) and inferred (red) dynamics of the latents $\boldsymbol{x}_t$ (B) and observations $\boldsymbol{y}_t$ (C) in the dynamic attractor model. Blue regions in B and C correspond to stimulation times. (D,E) True (black) and inferred (red) latent (D) and observation (E) dynamics shown in the 2D state space. Blue dots represent stimulated trajectories. Notice that the latents correspond to the polar coordinates of the observed trajectories and the observation model transforms latents from polar to Cartesian coordinates. (F) A synthetic trajectory generated by traversing a circle with constant speed. The inferred model captures the polar-to-Cartesian transformation without any prior knowledge only by using interventional data. (G,H) Comparison between SSM (observational model) and iSSM (ours). Reconstruction correlation between true and inferred latents (G) and observations (H) with increasing number of interventions are shown. With more interventions iSSM can better identify the latents. (I-P) Same as above for the Rotational Dynamics model of the motor cortex.

that whenever a neuron is perturbed, its activity is dissociated from all its upstream neurons. This assumption is easy to incorporate in a linear model, which is achieved by ignoring the columns in the dynamics matrix corresponding to the perturbed neuron. Denoting the interventional input to individual channels at time $t$ by $\boldsymbol{u}_t \in \mathbb{R}^M$, we model $\boldsymbol{x}, \boldsymbol{y}, \boldsymbol{u}$ as

$$\boldsymbol{x}_{t+1} = \mathbb{1}\{\boldsymbol{B}\boldsymbol{u}_t = 0\} \otimes \boldsymbol{A}\boldsymbol{x}_t + \boldsymbol{B}\boldsymbol{u}_t + \boldsymbol{\epsilon}_t, \quad \boldsymbol{y}_t \sim P(\boldsymbol{y}_t | f_{\boldsymbol{\theta}}(\boldsymbol{x}_t)).$$

where $\boldsymbol{\epsilon}_t \sim \mathcal{N}(\boldsymbol{0}, \boldsymbol{Q})$ and $\otimes$ denotes element-wise multiplication. $\boldsymbol{A} \in \mathbb{R}^{D \times D}$ captures spontaneous dynamics, while $\boldsymbol{B} \in \mathbb{R}^{D \times M}$ captures the effect of neural perturbations on latent dynamics. $\boldsymbol{Q} \in \mathbb{R}^{D \times D}$ is the covariance and $f_{\boldsymbol{\theta}}$ is a generic nonlinear function mapping latents to observations. If the intervention $\boldsymbol{u}_t$ is zero, the model follows spontaneous dynamics, but in the presence of an intervention, the model decouples the intervened node from its parents. While a non-interventional variant of this model was developed in prior work (Gao et al., 2016), our main contribution is to extend the model to the interventional regime and present theoretical results showcasing intriguing properties of the model. We term this variant of the model interventional SSM or iSSM for short.

## 3.2 INFERENCE

Since our model involves a nonlinear emission as well as non-conjugate noise model, we resort to variational inference techniques. Our goal is to infer the posterior distribution $p_{\boldsymbol{\theta}}(\boldsymbol{x}_{1:T} | \boldsymbol{y}_{1:T}, \boldsymbol{u}_{1:T})$ while optimizing the parameters $\boldsymbol{\theta}$. We follow the methodology of reparameterization and amortized inference but adapt some parts to our specific interventional scheme. For a review on variational methods for state space models see Archer et al. (2015). Denoting the approximate posterior distribution by $q_{\boldsymbol{\phi}}(\boldsymbol{x}_{1:T})$ the ELBO loss function is presented below:

$$\mathcal{L}(\boldsymbol{\phi}, \boldsymbol{\theta}) = \mathbb{E}_{\boldsymbol{\epsilon} \sim \mathcal{N}(\boldsymbol{0}, \boldsymbol{I})} \left[ \log p_{\boldsymbol{\theta}}(\boldsymbol{y}_{1:T}, \boldsymbol{u}_{1:T}, \boldsymbol{x}_{1:T}) - \log q_{\boldsymbol{\phi}}(\boldsymbol{x}_{1:T} | \boldsymbol{y}_{1:T}, \boldsymbol{u}_{1:T}) \right]$$

where $\boldsymbol{x}$ is reparameterized as $\boldsymbol{x}(\boldsymbol{\epsilon}) = \boldsymbol{\mu}_{\boldsymbol{\phi}} + \boldsymbol{\sigma}_{\boldsymbol{\phi}} \boldsymbol{\epsilon}$. The functions $\boldsymbol{\mu}, \boldsymbol{\sigma}$ are typically parameterized by neural networks (called recognition network) with an architecture that matches the dataset domain. Here we choose an LSTM for the recognition network.

Another important addition that makes the inference in our model possible is to apply the interventional structure directly in the approximate posterior during training. To do this, we replace $\boldsymbol{\mu}_t$ with

$1\{\boldsymbol{B}\boldsymbol{u}_t = 0\} \otimes \boldsymbol{\mu}_t + \boldsymbol{B}\boldsymbol{u}_t$ during each iteration of optimization. This ensures that the interventional data indeed manipulates the causal graph consistently in the approximate posterior.

### 3.3 THEORETICAL RESULTS: ON THE IDENTIFIABILITY OF iSSM

We provide sufficient conditions for the identifiability of iSSMs. We show that, given a sufficient set of *do*-interventions, one can identify both the latent dynamics matrix $\mathbf{A}$ and the mixing function $f_{\boldsymbol{\theta}}(\cdot)$ of the iSSM. This identifiability of the latent dynamics enables us to extrapolate to novel unseen interventions.

To identify the latent dynamics of iSSM, we proceed in three steps: (1) identify $P(\{f_{\boldsymbol{\theta}}(\boldsymbol{x}_t)\}_{t \in T})$ from the observed data distribution $P(\{\boldsymbol{y}_t\}_{t \in T})$; (2) identify $f_{\boldsymbol{\theta}}$ and $P(\{\boldsymbol{x}_t\}_{t \in T})$ from $P(\{f_{\boldsymbol{\theta}}(\boldsymbol{x}_t)\}_{t \in T})$ up to affine transformations; (3) further identify $f_{\boldsymbol{\theta}}$ and $P(\{\boldsymbol{x}_t\}_{t \in T})$ up to permutation, coordinate-wise shifting and scaling.

Begin with the first step of identifying $P(\{f_{\boldsymbol{\theta}}(\boldsymbol{x}_t)\}_{t \in T})$ from $P(\{\boldsymbol{y}_t\}_{t \in T})$. We make the following assumptions on the observation model.

**Assumption 3.1** (Bounded completeness of $P(\boldsymbol{y}_t|\boldsymbol{z}_t)$.)**.** The function $P(\boldsymbol{y}_t|\boldsymbol{z}_t)$—where $\boldsymbol{z}_t = f_{\boldsymbol{\theta}}(\boldsymbol{x}_t)$—is bounded complete in $\boldsymbol{y}_t$. Specifically, a function $f(X, Y)$ is bounded complete in $Y$ if $\int g(X)f(X, Y)\mathrm{d}X = 0$ implies $g(X) = 0$ almost surely for any measurable function $g(X)$ bounded in $L_1$-metric (Yang et al., 2017).

When the observational model satisfies the bounded completeness assumption, we can identify $P(\{f_{\boldsymbol{\theta}}(\boldsymbol{x}_t)\}_{t \in T})$ from $P(\{\boldsymbol{y}_t\}_{t \in T})$. (We detail the proof in Appendix A.) Many common functions $P(\boldsymbol{y}_t|\boldsymbol{z}_t)$ satisfy the bounded completeness condition, including exponential families (Newey & Powell, 2003), location-scale families (Hu & Shiu, 2018), and nonparametric regression models (Darolles et al., 2011). It is a common assumption to guarantee the existence and the uniqueness of solutions to integral equations, most commonly used in nonparametric causal identification in proxy variables and instrumental variables (Miao et al., 2018; Yang et al., 2017; D'Haultfoeuille, 2011). We refer the readers to Chen et al. (2014) for a detailed discussion of completeness.

We next proceed to identifying $f_{\boldsymbol{\theta}}$ and $P(\{\boldsymbol{x}_t\}_{t \in T})$ up to affine transformations. We require the following assumption on the mixing function $f_{\boldsymbol{\theta}}$.

**Assumption 3.2** (Mixing function)**.** The mixing function $f_{\boldsymbol{\theta}}(\cdot)$ is piecewise linear, continuous, and injective.

While the piecewise linear assumption may appear restrictive, we note that it entails flexible choices of $f_{\boldsymbol{\theta}}(\cdot)$, including (deep) ReLU networks that can approximate complicated functions.

We finally leverage the interventional data to achieve coordinate-wise identification of $f_{\boldsymbol{\theta}}$ and $P(\{\boldsymbol{x}_t\}_{t \in T})$. We make the following assumptions on the latent dynamics.

**Assumption 3.3** (No orphan latents)**.** There does not exist a non-zero vector $V$ such that $Cov(V^\top \boldsymbol{x}_{t+1}, V^\top \boldsymbol{x}_t) = 0$ for all $t$.

Loosely, this assumption guarantees that no latent dimension in $\boldsymbol{x}_t$ is an orphan node, namely a node that is never affected by itself nor by other nodes. In other words, each latent has at least one (non-trivial) causal parent from the previous timestep.

We further describe the requirements of the interventions that needs to be performed for identifying iSSM.

**Assumption 3.4** (do-interventions on each latent node)**.** There is at least one $\mathrm{do}$-intervention (i.e. non-random $\boldsymbol{u}_t$) being performed on each latent dimension of $\boldsymbol{x}_t$.

Assumption 3.4 requires the interventions be $\mathrm{do}$-interventions, which would break all the connections between some component—$x_{t+1,j}$ for some $j$—and its causal parents $\boldsymbol{x}_t$. The $\mathrm{do}$-interventions thus induce the statistical independence between the intervened variables over time. This independence is the crucial signature we leverage to identify the latent $\boldsymbol{x}_t$ and the mixing function $f_{\boldsymbol{\theta}}(\cdot)$.

Under these assumptions, we can achieve the identification of iSSM as follows.

**Theorem 3.5** (Identifiability of iSSM)**.** *Under Assumptions 3.1 to 3.4, the latent dynamics $\boldsymbol{A}$ and the mixing function of $f_{\boldsymbol{\theta}}(\cdot)$ can be identified up to permutation, and coordinate-wise shifting and*

scaling, namely $\hat{\boldsymbol{A}} = \boldsymbol{A}\Lambda\Pi + \boldsymbol{c}$, where $\Lambda$ is an invertible diagonal matrix, $\Pi$ is a permutation matrix, and $\boldsymbol{c}$ is a constant vector. As a consequence, one can also identify the observations' distribution $P(\{\boldsymbol{y}_t\}_{t \in T})$ under novel unseen $\boldsymbol{u}_t$ interventions.

The proof of Theorem 3.5 is in Appendix A. This result establishes the identifiability of iSSM and its predictive power for unseen interventions. Moreover, it illustrates how interventions can help identify latent variables via inducing statistical independence among the latents, revealing latent dynamics in non-linear state-space models.

## 4 RESULTS

### 4.1 IDENTIFYING MOTOR CORTICAL DYNAMICS IN SIMULATIONS

To illustrate how iSSM leads to identification, we take inspiration from models of motor cortex. A key observation in the motor cortex made by multiple groups is the presence of rotational dynamics (Churchland et al., 2012). From a computational perspective, it has been argued that rotational dynamics provide a basis for motor neuron activations and muscle movements. It has been argued that rotational basis provides robustness to noise and interventions (Logiaco et al., 2021). Inspired by these observations and results, multiple dynamical models for the rotational activities in the motor cortex have been proposed (Laje & Buonomano, 2013; Sussillo et al., 2015). The first model, called *Rotational Dynamics (RD)* proposes that the motor cortex has underlying rotational dynamics. As a result, in this model the rotational dynamics are generated within the motor cortex independent of input or feedback activity (Fig. 2I; Sussillo et al. (2015)). Eq. 1 describes the dynamics and emissions of *RD*.

$$\textbf{Rotational Dynamics:} \quad \frac{d\boldsymbol{x}}{dt} = \begin{bmatrix} 0 \\ a\boldsymbol{x}_1 \end{bmatrix} + \boldsymbol{\epsilon}_t, \quad \boldsymbol{y}_t = \begin{bmatrix} \boldsymbol{x}_1 \cos(\boldsymbol{x}_2) \\ \boldsymbol{x}_2 \sin(\boldsymbol{x}_2) \end{bmatrix} + \boldsymbol{\delta}_t \qquad (1)$$

The second model, called *Dynamic Attractor (DA)* assumes that the underlying dynamics of the motor cortex is a rounded attractor. In this model, the rotational dynamics in motor neurons are generated by some upstream region moving the state along the attractor (Laje & Buonomano, 2013). Eq. 2 describes the dynamics and emissions of *DA*.

$$\textbf{Dynamic Attractor:} \quad \frac{d\boldsymbol{x}}{dt} = \begin{bmatrix} a_1 \boldsymbol{x}_1 \\ a_2(1 - \boldsymbol{x}_2) \end{bmatrix} + \boldsymbol{\epsilon}_t, \quad \boldsymbol{y}_t = \begin{bmatrix} (1 - \boldsymbol{x}_1)\cos(\boldsymbol{x}_2) \\ (1 - \boldsymbol{x}_2)\sin(\boldsymbol{x}_2) \end{bmatrix} + \boldsymbol{\delta}_t \qquad (2)$$

While these models have distinct characteristics and propose different underlying circuit mechanisms, Galgali et al. (2023) show that the trial averages of these models can be precisely the same, limiting our ability to identify the true dynamics of the motor cortex solely from observational data.

O'Shea et al. (2022) refer to these models as low-dimensional vs. path-following dynamical systems and use an interventional strategy to discover whether the dynamics in the motor cortex follows either of these regimes. Similarly, here we ask if interventional data can distinguish between these models. To address this, we generate data from *RD* and *DA*. The latent states $\boldsymbol{x}(t)$ in both models follows linear dynamics, while the observation model in both cases is highly nonlinear. Therefore, recovering the true latents is not a trivial task. During data generation, We apply repeated interventions interleaved by resting periods for the network to go back to its stationary state. The dynamics of latents and observations are shown in Fig. 2B-E,J-M. While in the absence of interventions both models produce the same trajectories, one can observe that interventional trajectories exhibit distinct characteristics (Fig. 2E,M).

Consistent with O'Shea et al. (2022) our results suggest that in the presence of interventional data using the iSSM model one can identify the underlying dynamics and emissions (Fig. 2F,N) and recover the true latent variables (Fig. 2G,O). This recovery keep improving as we collect more interventional data emphasizing the importance of perturbation experiments in causal hypothesis testing (Fig. 2G,O).

### 4.2 IDENTIFYING DYNAMICS UNDERLYING SHORT-TERM MEMORY IN MICE

Persistent activity is a hallmark of short-term memory across species (Romo et al., 1999; Fuster & Alexander, 1971). How can a network of neurons produce activities in response to an input stimulus

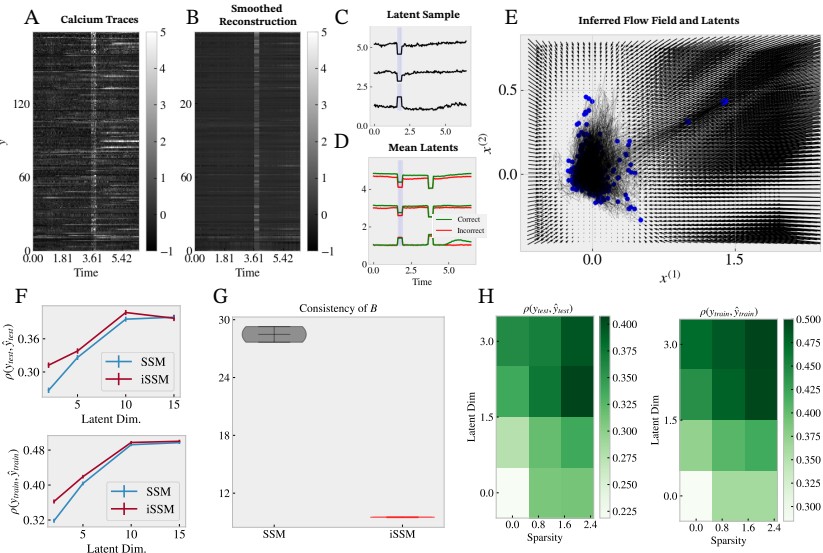

Figure 3: **Results on Mice Dataset.** (A) Calcium responses of ALM neurons during stimulation. The white high-activity band corresponds to the stimulation. (B) Smoothed responses given by the mean posterior of the model. (C) Latents discovered by the model shown for one trial. The blue bands correspond to stimulation times. (D) Mean latents for correct vs. incorrect trials. The dynamics of our identifiable latents distinguish between correct and incorrect trials without any prior knowledge of the behavior. (E) Flow field inferred by our model shows a slow attractor on the left. When the state is perturbed to the top right the dynamics quickly push it back to the attractor suggesting low-dimensional dynamics. (F) Correlation between true and inferred observations for the test (top) and train (bottom) sessions when with increasing number of latents for both SSM and iSSM models. (G) The $B$ matrices are consistent across random initializations of the model only for iSSM and not for SSM. (H) Test (left) and train (right) reconstruction accuracy for iSSM as a function of number of latents and the sparsity parameter for the $B$ matrix. Both higher sparsity and larger number of latents improve the accuracy.

that are maintained after the stimulus is removed? Multiple network mechanisms are proposed to underlie persistent activity. Among those, one popular model is known as *Functionally Feedforward (FF)* model (Goldman, 2009). *FF* assumes that the network constitutes of a few smaller subnetworks that are connected to each other in feedforward manner. Since these subnetworks do not necessarily need to form a spatial cluster in the brain, experimentally finding footprints of this type of connectivity is not feasible. However, theoretical properties of the model has been well-studied. For example, it is commonly argued that *FF* results in robustness to structural noise (Qian et al., 2024). An alternative model for the persistent activity is known as *Line Attractor (LA)* model (Seung, 1996). Under *LA* circuit model, the activity of an upstream region pushes the state of the circuit along the line attractor, and the dynamics preserves the state until a new input is arrived.

Various sources of non-identifiability make it challenging to recover the true latents and dynamical mechanisms. We elaborate on two of these sources here.

First, neural recordings are undersampled, meaning that from a large pool of neurons involved in the computation only a small fraction are recorded. Undersampling (also known as partial observation) is indeed a significant origin of non-identifiability discussed in the literature (Beiran & Litwin-Kumar, 2024). A recent theoretical study investigates the effect of undersampling specifically in the context of persistent activity (Qian et al., 2024). They show that when the network is undersampled, observational models have a built-in bias for characterizing the model as *FF* regardless of the underlying mechanism (Qian et al., 2024).

Second, non-identifiability can be caused by the mixing of input-driven and recurrent activity in the network (see Galgali et al. (2023) for a more detailed discussion). Lipshutz et al. show that noise correlations can be used to disentangle input-driven and recurrent activity. Note that noise

correlations can be considered as small perturbations around the mean trajectory. Hence, consistent with Lipshutz et al. our results suggest that interventions are necessary to distinguish between these hypotheses.

We applied iSSM to a public dataset of targeted photostimulation in the anterior lateral motor cortex (ALM) of mice during a short-term memory task (Daie et al., 2021). The task included a sample epoch where an auditory cue guided the mice for a left vs right cue to get water reward. The sample epoch was followed by a delay epoch of 3 seconds where the mice needed to engage working memory to keep track of the guided cue. Finally during the response period the mice received the reward if the lick direction was correct. The photostimulation was delivered during the delay period for a short amount of time started simultaneously with the delay period or after 1 or 2 seconds.

Calcium recordings were done in 179 identified neurons for 77 repeated trials 3A. There were 8 photostimulation channels targeted to stimulate neurons according to their response selectivity. We run the model using a latent dimension of 3 for visualization purposes. We set the dimension of interventional inputs $u_t$ to the number of photostimulation channels and fitted the stimulation matrix $B$ with a sparsity penalty. The smoothed and denoised neural activities are shown in Fig. 3B. The reconstruction accuracy of the data for both training and testing trials (Fig. 3F) were larger than the baseline SSM model across a range of hyperparameters (Fig. 3F,H). Furthermore, the latents learned by the model show distinct mean trajectories for correct vs. incorrect trials suggesting that they capture behaviorally meaningful dynamics (Fig. 3C,D). Finally, we present the flow-field fitted by the model which shows a slow mode for observational data and a fast mode for interventional data pushing the state back towards the attractor (Fig. 3G).

A hallmark of identification is robustness to initialization. To test whether iSSM results in identifiable latents, we ran the model several times with different random initializations and inferred the latents as well as the stimulation matrix $B$. In Fig. 3G we show the consistency of the inferred $B$ matrix across different seeds. The consistency is computed by first aligning the columns of the $B$ matrix to account for permutation invariance of the latents, followed by computing the Euclidean distance between aligned $B$ matrices. The aligned distances are considerably smaller for iSSM compared to SSM providing evidence for the identifiability.

### 4.3 GENERALIZING TO NEW INTERVENTIONS IN MACAQUE MONKEYS

Understanding network dynamics to control behavior has been a longstanding challenge in neuroscience. The overarching goal is to deliver targeted stimulation to a network of neurons to steer the dynamics or the behavior towards a pre-determined outcome (Haimerl et al., 2023; Jou et al., 2023). A first step towards understanding the circuit effects or behavioral influences of network manipulations is to build models that can predict the response to interventions. The space of possible interventions is combinatorial and intractable to cover. Therefore, an alternative approach is to build models that can generalize to unseen interventions.

We showed theoretically in section 3.3 that iSSM has this property. Concretely, if we fit the iSSM model to a interventional data, where the dataset consists of a small set of canonical interventions, the model is able to generalize to unobserved interventions. To validate this empirically, we showed results on a synthetic datasets (Fig. 2). Here we want to test whether these results hold in a real biological dataset.

The dataset consisted of electrophysiological recordings using electrode arrays implanted on the prefrontal cortex of macaque monkeys during quiet wakefulness (resting) while the animals were sitting awake in the dark. The electrode array included 96 electrodes that were also used for delivering micro-circuit electrical stimulations (Nejatbakhsh et al., 2023). We analyzed 6 datasets, 3 with only observational data and 3 with a combination of observational and interventional data.

In Fig. 4A,D we show firing rates recorded from each of the 96 electrodes for an interventional (Fig. 4A; the vertical white bars correspond to stimulation times) and observational (Fig. 4D) session. In each interventional session, two electrodes were repeatedly stimulated while recordings were performed from all other electrodes. We fit the iSSM model with a latent dimension of $D = 2$ and use it to denoise the data (Fig. 4B,E). The inferred flow fields for an interventional and observational session are shown in Fig. 4D,F respectively. The stimulation matrix $B$ is depicted in Fig. 4I showing that some electrodes have excitatory and other electrodes have inhibitory causal effects on the latents.

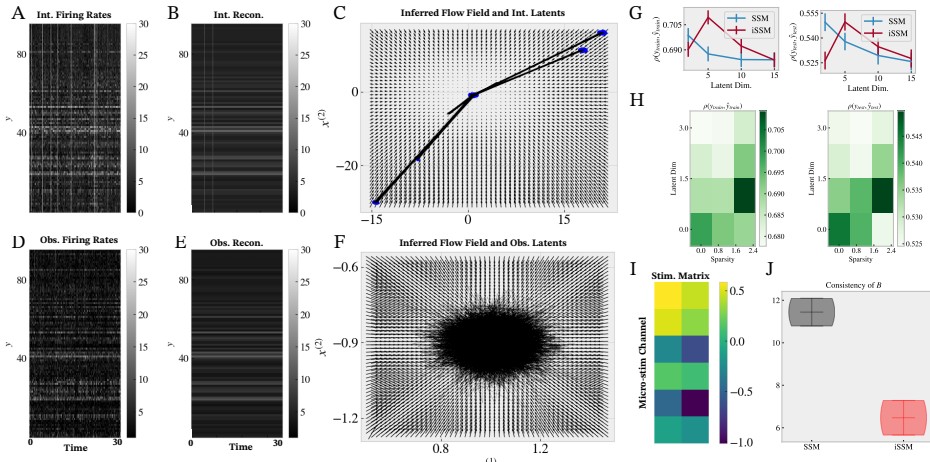

Figure 4: **Results on Monkey Dataset.** (A) Unit responses for a training interventional session. (B) Inferred smooth responses for the same trial in A. (C) Flow field inferred by the model shows attractor like structure. (D-F) Same as (A-C) for a test observational trial. (G) Comparison between SSM and iSSM with increasing number of latents for train (left) and test (right) reconstruction accuracy. Both SSM and iSSM benefit from larger number of latents with iSSM consistently outperforming SSM. (H) Train (left) and test (right) reconstruction accuracy is shown with varying number of latents and sparsity parameter for $B$ matrix. In this dataset and intermediate number of latents is desired. (I) Matrix $B$ inferred by the model showing the effect of stimulating each unit (rows) on each latent (columns). Some neurons have inhibitory effect on the latent and some have excitatory effect. (J) Consistency of the inferred $B$ matrix across random initializations only for iSSM and not for SSM.

The reconstruction accuracy on the training and testing session are larger for iSSM compared to baseline SSM across a range of hyperparameters, suggesting that the model can better generalize to unseen sessions (Fig. 4G,H).

## 5 DISCUSSION

### 5.1 SUMMARY

Here we proposed iSSM, a framework for joint modeling of observational and interventional data. We provided theoretical results showing that iSSM model when fitted on interventional data leads to identifiability of latents as well as dynamics and emissions.

To illustrate iSSM's applicability, we showed results on 3 different examples covering a range of assumptions. The first example was a synthetic dataset with linear dynamics and nonlinear emissions. The second example was calcium recordings from mouse ALM region with targeted photostimulation delivered by channels that did targeted groups of neurons. The third example was electrophysiological recordings from macaque monkey prefrontal cortex with micro-stimulation delivered by the same recording electrodes. In all cases, our results show impressive generalization capabilities and parameter recovery suggesting that when models that are theoretically grounded are applied to interventional data they are capable of testing sophisticated causal hypotheses.

### 5.2 LIMITATIONS

In this work, we focused on a generative model that has linear dynamics. While the inference model can still capture nonlinearities through its recognition network, explicitly modeling nonlinearities and providing theoretical results is an important limitation of this work. In addition, our results on biological datasets are mostly exploratory and further validation experiments are required to confirm these results. We leave these for future work.

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

## A   PROOF OF THEOREM 3.5

We consider the interventional state space model (iSSM),

$$\boldsymbol{y}_t \sim P(\boldsymbol{y}_t | f_{\boldsymbol{\theta}}(\boldsymbol{x}_t)), \tag{3}$$

$$\boldsymbol{x}_{t+1} = 1\{\boldsymbol{B}\boldsymbol{u}_t = 0\} \otimes \boldsymbol{A}\boldsymbol{x}_t + \boldsymbol{B}\boldsymbol{u}_t + \boldsymbol{\epsilon}_t. \tag{4}$$

**Step I: Identifying the distribution of $\boldsymbol{z}_t \triangleq f_{\boldsymbol{\theta}}(\boldsymbol{x}_t)$.**   We begin with identifying the marginal distribution of $P(\boldsymbol{z}_t)$ from $P(\boldsymbol{y}_t)$. The core assumption we rely on in this step is bounded completeness, which we define in Assumption 3.1

The bounded completeness of $P(\boldsymbol{y}_t | \boldsymbol{z}_t)$ implies that $P(\boldsymbol{z}_t)$ is identifiable from $P(\boldsymbol{y}_t)$. It is because $P(\boldsymbol{z}_t)$ must be the unique solution to the integral equation $\int P(\boldsymbol{y}_t | \boldsymbol{z}_t) P(\boldsymbol{z}_t) \mathrm{d}\boldsymbol{z}_t = P(\boldsymbol{y}_t)$. Specifically, if there are two solutions to this equation $\hat{P}_1(\boldsymbol{z}_t), \hat{P}_2(\boldsymbol{z}_t)$, then they must be equal. It is due to the bounded completeness of $P(\boldsymbol{y}_t | \boldsymbol{z}_t)$: the two solutions must satisfy $int P(\boldsymbol{y}_t | \boldsymbol{z}_t)[\hat{P}_1(\boldsymbol{z}_t) - \hat{P}_2(\boldsymbol{z}_t)] \mathrm{d}\boldsymbol{z}_t = 0$, which implies $\hat{P}_1(\boldsymbol{z}_t) = \hat{P}_2(\boldsymbol{z}_t)$.

**Step 2: Affine identification of $f_{\boldsymbol{\theta}}(\cdot)$ and $P(\{\hat{\boldsymbol{x}}_t\}_{t \in T})$.**   In this step, we establish the affine identification of the mixing function $f_{\boldsymbol{\theta}}(\cdot)$ by invoking Theorem 3.5 of Balsells-Rodas et al. (2023): identifying $f_{\boldsymbol{\theta}}(\cdot)$ from $P(f_{\theta}(\boldsymbol{x}_t))$ is a special case of identifying the mixing function in a switching dynamical system.

To enable identification, we require Assumption 3.2. In particular, the mixing function should be a piece-wise linear function.

**Lemma A.1** (Theorem 3.5 of Balsells-Rodas et al. (2023))**.** *Under Assumption 3.2, the mixing function $f_{\boldsymbol{\theta}}(\cdot)$ and the latent distribution $P(\{\hat{\boldsymbol{x}}_t\}_{t \in T})$ can be identified from $P(f_{\theta}(\boldsymbol{x}_t))$ up to affine transformation.*

This lemma is an instantiation of Theorem 3.5 in Balsells-Rodas et al. (2023) in the special case of linear transition dynamics.

**Step 3: Identification of $\boldsymbol{x}_t$ via interventions.**   The previous step shows that we can identify $\boldsymbol{x}_t$ up to affine transformation. In this step, we show that, if two solutions of $\boldsymbol{x}_t$ are affine transformations of each other, they must coincide if they agree on the interventional distributions, under Assumptions 3.3 and 3.4. This argument implies that the interventional distributions can identify $\boldsymbol{x}_t$ (up to permutation, and coordinate-wise shifting and scaling.)

Concretely, consider two sets of latent variables $\{\boldsymbol{x}_t\}_{t \in T}$ and $\{\hat{\boldsymbol{x}}_t\}_{t \in T}$ where they are affine transformations of each other

$$\hat{\boldsymbol{x}}_t = M\boldsymbol{x}_t + c, \forall t. \tag{5}$$

Suppose both sets satisfy Equation (4) across all intervention environments, namely,

$$\boldsymbol{x}_{t+1} = 1\{\boldsymbol{B}\boldsymbol{u}_t = 0\} \otimes \boldsymbol{A}\boldsymbol{x}_t + \boldsymbol{B}\boldsymbol{u}_t + \boldsymbol{\epsilon}_t, \tag{6}$$

$$\hat{\boldsymbol{x}}_{t+1} = 1\{\hat{\boldsymbol{B}}\boldsymbol{u}_t = 0\} \otimes \hat{\boldsymbol{A}}\hat{\boldsymbol{x}}_t + \hat{\boldsymbol{B}}\boldsymbol{u}_t + \hat{\boldsymbol{\epsilon}}_t, \tag{7}$$

where both $\boldsymbol{\epsilon}_t, \hat{\boldsymbol{\epsilon}}_t$ are i.i.d over time. Then we will prove that $M = \Lambda \Pi$, where $\Lambda$ is an invertible diagonal matrix, and $\Pi$ is a permutation matrix.

We achieve identification using the following observation. Suppose the $j$th latent $x_{t,j}$ was intervened in an environment, namely $1\{(B\boldsymbol{u}_t)_j = 0\} = 0$. Then we have

$$x_{t,j} = (\boldsymbol{B}\boldsymbol{u}_t)_j + \epsilon_{t,j} \qquad \forall t, \tag{8}$$

and thus $x_{t+1,j} \perp \boldsymbol{x}_t$ for all $t$. The reason is that the intervention set $\mathbf{x}_{t+1,j}$ to be $(\boldsymbol{B}\boldsymbol{u}_t)_j$ plus a random noise component, hence independent of all components of $\mathbf{x}_t$.

Below we argue that, if we also find a component $j'$ of $\hat{\boldsymbol{x}}_{t+1}$ such that $\hat{x}_{t+1,j'} \perp \hat{\boldsymbol{x}}_t$, then $M_{j',-j} = 0$, i.e. $\hat{x}_{t+1,j'}$ must be an affine transformation of $x_{t+1,j}$.

To make this argument, we write

$$\hat{\boldsymbol{x}}_{t+1,j'} = M_{j',-j}^\top \boldsymbol{x}_{t+1,-j} + M_{j',j} x_{t+1,j} + c_{j'}, \tag{9}$$

$$\hat{\boldsymbol{x}}_{t,j'} = M_{j',-j}^\top \boldsymbol{x}_{t,-j} + M_{j',j} x_{t,j} + c_{j'}. \tag{10}$$

Then since $\hat{x}_{t+1,j'} \perp \hat{\boldsymbol{x}}_t$, we have that

$$Cov(\hat{x}_{t+1,j'}, \hat{\boldsymbol{x}}_t) = 0. \tag{11}$$

This implies

$$
\begin{align}
0 =& Cov(\hat{x}_{t+1,j'}, \hat{\boldsymbol{x}}_t) \tag{12} \\
=& Cov(M_{j',-j}^\top \boldsymbol{x}_{t+1,-j} + M_{j',j} x_{t+1,j}, M_{j',-j}^\top \boldsymbol{x}_{t,-j} + M_{j',j} x_{t,j}) \tag{13} \\
=& Cov(M_{j',-j}^\top \boldsymbol{x}_{t+1,-j}, M_{j',-j}^\top \boldsymbol{x}_{t,-j}) + Cov(M_{j',-j}^\top \boldsymbol{x}_{t+1,-j}, M_{j',j} x_{t,j}) \\
& + Cov(M_{j',j} x_{t+1,j}, M_{j',-j}^\top \boldsymbol{x}_{t,-j}) + Cov(M_{j',j} x_{t+1,j}, M_{j',j} x_{t,j}) \tag{14} \\
=& Cov(M_{j',-j}^\top \boldsymbol{x}_{t+1,-j}, M_{j',-j}^\top \boldsymbol{x}_{t,-j}). \tag{15}
\end{align}
$$

The last equation is due to Equation (8). It implies that $M_{j',-j} = 0$ due to Assumption 3.3. In other words, the $j'$th dimension of $\hat{\boldsymbol{x}}_t$ that achieves the independence property is mapped to the $j$th dimension of $\boldsymbol{x}_t$ up to scaling and shifting; one can separate out the intervened latent from the unintervened ones up to permutation, and coordinate-wise shifting and scaling.

Repeating this argument with intervention data on all other latents (Assumption 3.4), we can identify the whole set of latents up to permutation, and coordinate-wise shifting and scaling, namely $M = \Lambda\Pi$.

Identifying the latents up to permutation, and coordinate-wise shifting and scaling implies that one can identify the latent dynamics matrix $\boldsymbol{A}$ also up to permutation, and coordinate-wise shifting and scaling.

Finally, as a consequence of identifying all parameters of the iSSM, we can predict the observation distributions for novel unseen interventions $\boldsymbol{u}_t$.

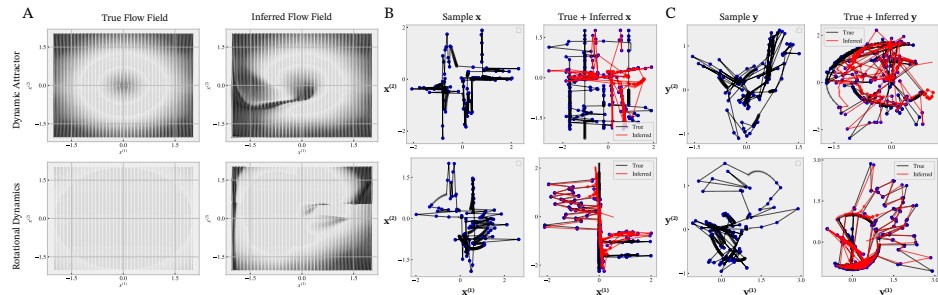

Figure 5: **Supplementary Results on Models of Motor Dynamics.** (A) True (left) vs. inferred (right) flow fields for *DA* and *RT* models of the motor system. (B) Latent samples from the generative model. While the generative samples possess the qualitative features of the models samples from the recognition model better capture the data. (C) Observation samples from the generative model.

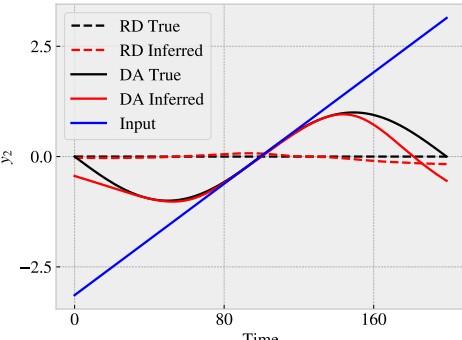

Figure 6: **Testing Between the Two Models.** To test whether the iSSM model enables testing between *DA* and *RD* we input both recognition models with the signal shown in blue and generate trajectories (we only show the second dimension of the input and observations for clarity). We expect the *DA* model to generate a sinusoidal while we expect *RD* to stay close to zero. This result shows that the recognition models are indeed sufficient for testing the two hypotheses.

## B  EXPERIMENTAL DETAILS

In this section we present additional detail on the models of motor dynamics (Fig.5,6). Furthermore, present a new application of iSSM in uncovering mechanisms of working memory in simulations (Fig. 7).

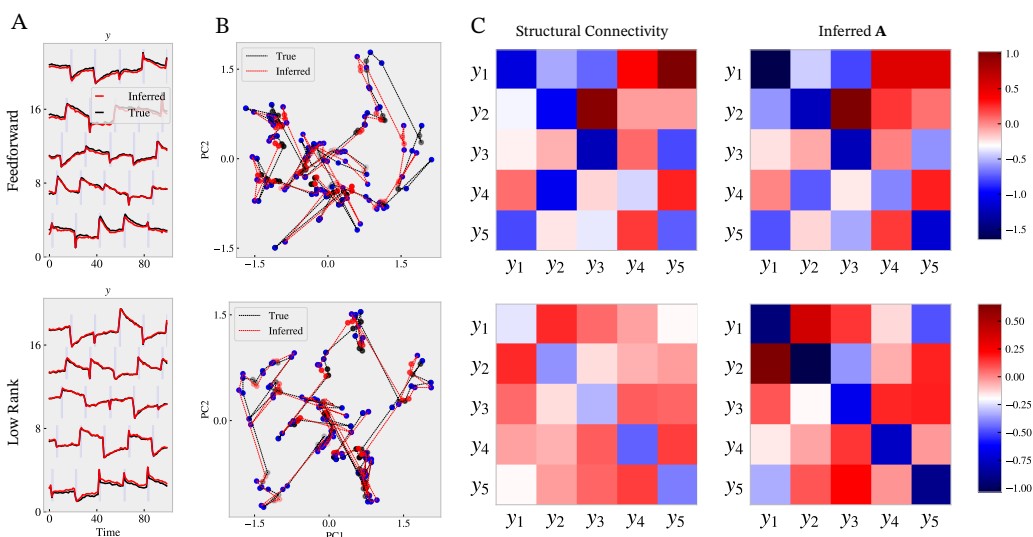

Figure 7: **Models of Working Memory.** Following Qian et al. (2024) here we generate data from feedforward (*FF*) and low-rank (*LR*) models of working memory to test whether iSSM can recover the true underlying flow field parameterized by the structural connectivity matrix. (A) Signals generated from *FF* (top) and *LR* (bottom) in 5 dimensions. (B) Same data shown in the top 2 PC space. (C) True dynamics matrix (or structural connectivity) of the models are shown on the left. iSSM recovers the main characteristic features of these matrices and enables distinguishing between the two models of working memory.

