# OpenReview forum: "Identifying neural dynamics using interventional state space models"
_ICLR.cc/2025/Conference — ICLR 2025 Conference Withdrawn Submission_

### Official Review · Reviewer_RAGV · 2024-10-31

**Soundness:** 2
**Presentation:** 2
**Contribution:** 2
**Rating:** 3
**Confidence:** 5

**Summary:**

The authors address the problem of constructing identifiable models of neural population activity by developing interventional state-space models. The idea is to utilize the theory of causal inference, which comes equipped with protocols and assumptions for model identification. The authors develop the model, prove identifiability under standard causal inference assumptions, and perform numerical investigations in three contexts with interventions: synthetic data, motor cortex Ca imaging, prefrontal cortex electrophysiology.
Unfortunately, as outlined below, this paper has several limitations that prevent acceptance in its current form.

**Strengths:**

The theory is clear and well written, but is a straightforward combination of existing analytic approaches.

**Weaknesses:**

The most fatal flaws center around the mismatch between a key assumption of the theory/models (3.4) and the actual experimental data context that is targeted, lack of any examination, in either simulated or real data, on the impact of deviations from the assumption, combined with the complete lack of comparison to other models.
In more detail–
The theory is clear and well written, but is a straightforward combination of existing analytic approaches.
-Assumption 3.4 of causal inference is a very strong assumption, and it is not clear how to provide perturbation in systems neuroscience that adhere to it, and it is not discussed how to check this assumption in data. Indeed, it appears naively that the data displayed in Fig.3A,Fig.4A, in which the perturbation looks to affect most neurons, directly violates this assumption. This is a fatal flaw in the stated context of the manuscript, as the entire motivation for the paper and numerical validation are in the context of neuroscience.
-only low-d latent spaces are considered, but data could be high-d depending on task
-relatedly, selection of number of dimensions is not discussed
-compute time of algorithm not discussed? Useful in real-time?
-quantification:
 –corr.coef is used, and the values look very low to me (e.g., Fig.3c,d, Fig.4I), so while there may be differences from 0, whether or not the model is meaningfully predictive is brought into doubt by the small effect size.
– samples sizes are not clear
    –statements about significance are vague (e.g., pg. 8 first paragraph)
-visuals are very small, hard to read, and poorly explained
-lots of plots, the relevance or results of which are poorly explained
	-there is no comparison to other methods whatsoever. Given the issues with
Assumption 3.4 discussed above, this constitutes a fatal flaw in the current context, as the theory can not be used to justify the algorithm in the data to which it is applied, and as such other models/algorithms could perform comparably.
-no software availability as far as I could tell.
-no validation on non-interventional data (as far as i can tell) as another benchmark
-no numerical assessment of identifiability in the data. This could have been done in a number of ways, such as looking at the variance of the inferred variables in real data, or investigating how deviations from Assumption 3.4 are evidenced (e.g., systematic errors or magnitude of deviations across parameters) in the solutions.

**Questions:**

n/a

---

> ### Author Response · Authors · 2024-11-29
> **Response**
>
> > The theory is clear and well written, but is a straightforward combination of existing analytic approaches.
>
> In the first paragraph of our general response we’ve tried to address this concern.
>
> > Assumption 3.4 of causal inference is a very strong assumption, and it is not clear how to provide perturbation in systems neuroscience that adhere to it, and it is not discussed how to check this assumption in data.
>
> We’ve included a discussion on this point in our general responses under section **4** and **5** in our general response; we hope it can partially address this comment.
>
> > Only low-d latent spaces are considered, but data could be high-d depending on task.
>
> We’ve added new experiments with varying dimensions in Fig 3 and 4. As we discuss in the general response the optimal dimension depends on the amount of data and number of interventions. With few interventions on a few dimensions we cannot hope for recovering all the dimensions of true latents, but we can identify the system for the dimensions that we’ve observed data.
>
> > Relatedly, selection of number of dimensions is not discussed.
>
> To address this, we’ve performed new cross-validation results in the revised manuscript. Please see our general response under section **2**.
>
> > Compute time of algorithm not discussed? Useful in real-time?
>
> The algorithm is linear in time and the number of dimensions. The algorithmic complexity follows the best practices in variational inference for SSMs which is quite tractable. Our codes are implemented in *jax* which is compatible with GPU and our experiments took mostly a few minutes to run.
>
> > Corr. coeff is used, and the values look very low to me.
>
> We have included comparisons against SSM to have a baseline to compare against (section **1** of our general response). In addition, we replaced the error bars to represent standard deviation (std) as opposed to standard error of the mean (sem) to make the comparisons clearer. Regarding the visuals, we will improve the font sizes and other aesthetics for the camera-ready paper if we get accepted.
>
> > Lots of plots, the relevance or results of which are poorly explained.
>
> We would appreciate it if you point us to specific instances where we can focus on.
>
> > There is no comparison to other methods whatsoever.
>
> Thank you for the suggestion, we agree that we’ve been missing a critical comparison which is the comparison against SSM, which we’ve now included in the revised manuscript. Please also see section **1** of our general response.
>
> > No software availability as far as I could tell.
>
> We have a code repository which is implemented according to best practices and standards of academic software development. We will publicly release the code package upon acceptance. Here, we include an [anonymized version of the code package](https://drive.google.com/file/d/1Rqw2wSIdIsRARo6puf0bxGmh5jEX9CVo/view?usp=sharing) for your consideration.
>
>
> > No validation on non-interventional data (as far as i can tell) as another benchmark.
>
> We agree, please see our earlier response to your comment on comparisons and section **1** of our general response.
>
> > No numerical assessment of identifiability in the data.
>
> This is a great suggestion, for experiments presented in Fig. 3, 4 we’ve run the model multiple times with random initialization and calculated the variance of the (aligned) stimulation matrices $\boldsymbol{B}$ across seeds. Since our latents and inferred $\boldsymbol{B}$ matrices are permutation invariant, the alignment procedure finds the best permutation that matches the columns of $\boldsymbol{B}$ across different runs. Our results suggest that iSSM’s inferred parameters are robust across different runs while SSM does not have this property. We’ve included these results in Fig. 3G and Fig. 4J.

---

### Official Review · Reviewer_GFUM · 2024-10-31

**Soundness:** 3
**Presentation:** 3
**Contribution:** 2
**Rating:** 5
**Confidence:** 4

**Summary:**

This paper proposed interventional state space models to causally predict neural responses to novel perturbations. The authors build a multiplication to the latent space to encode the interventional input. The model is tested on simulated data and is able to reconstruct the true latent and observations. The model is then tested on real data and is able to generalize to unseen interventional input.

**Strengths:**

The proposed iSSM jointly models the observational and interventional data, the model is applied to three different datasets and shows efficiency. The model can generalize to new interventions in the third dataset.

**Weaknesses:**

Lack of comparison: the authors mentioned that their contribution is to extend the model to the interventional regime, however, the authors did not show how this added interventional input affects the reconstruction or other aspects.

**Questions:**

1.	What do those blue dots mean in Figure 2 D and E?
2.	What do the inferred flow fields look like in Figure 2?
3.	How do you define input $u$? Also, in your simulation in section 4.1, are there any inputs?
4.	How do you choose your parameters? For example, dimension of latent. Did you try higher dimensions?
5.	In the first equation on page 4, does that mean when B$u_t$=0, 1 is multiplied to A$x_t$? Then what about B$u_t$ is not 0? You mentioned that the model decouples the intervened node from its parents, but the equation did not show this.
6.	Some equations are not labeled.

---

> ### Author Response · Authors · 2024-11-29
> **Response**
>
> > Lack of comparison: the authors
>
> Thank you for pointing this out, please see section **1** of our general response.
>
> > What do those blue dots mean ...
>
> The blue dots correspond to interventions. They represent the state changes when the intervention is applied to the system. We’ve added this to the figure caption for clarity.
>
> > What do the inferred flow fields look like in Figure 2?
>
> This is an interesting question. There are two ways for computing the flow fields, (1) Using the generative model where the flow fields are given by the linear system that governs our latent dynamics followed by the emission model (2) Using the recognition model to infer the latents followed by the emission transform. The flow fields computed based on (2) indeed resemble the true flow fields as illustrated in Fig. 5 in the appendix. The discrepancy between (1) and (2) could be due to optimization, the mismatch between generative and inference function families, and the ELBO gap. This is a common issue for all statistical models that use variational inference [1,2].  To investigate this gap, we’ve included trajectories generated by the generative and recognition models in Fig. 5 of the appendix.
>
> > How do you define input u?
>
> We denote the interventional inputs by $\boldsymbol{u}$. Normally the magnitude of $\boldsymbol{u}$ is determined by the experimentalist. In our case we’ve set the magnitude to be in the same order as the maximum value of neural responses during interventions. Notice that the magnitude change can be recovered by $\boldsymbol{A}$ and $\boldsymbol{B}$ matrices making the model identifiable up to re-scaling.
>
> > How do you choose your parameters?
>
> We’ve added new cross-validation experiments addressing this. Please see section **2** of our general response.
>
>
> > In the first equation on page 4, does
>
> Let’s consider a simple case where $\boldsymbol{B}$ is equal to identity, and the interventions are performed on individual neurons (therefore the latent and observed dimensions are the same $D=N$). When there’s an interventional input in a certain dimension of $\boldsymbol{u}$ say $\boldsymbol{u}^{(i)}$ that makes $\boldsymbol{B}\boldsymbol{u}^{(i)}$ non-zero at that time point which makes the corresponding dimension in $\boldsymbol{A}\boldsymbol{x}_{t-1}$ to zero. Therefore the latents at time $t$ is given by $\boldsymbol{x}^{(i)}_t = \boldsymbol{u}^{(i)}_t$ which means that the interventional input replaces the activity at time $t$ and $\boldsymbol{x}^{(i)}$ is decoupled from its parents in the network. We hope this clarifies our argument, let us know if further clarification is needed.
>
> > Some equations are not labeled.
>
> Thank you for pointing this out, we will fix this in the camera-ready paper upon acceptance.
>
>
> **References**
>
> [1] Yao, Yuling, et al. "Yes, but did it work?: Evaluating variational inference." International Conference on Machine Learning. PMLR, 2018.
>
> [2] Yao, Yuling, and Justin Domke. "Discriminative calibration: check bayesian computation from simulations and flexible classifier." Advances in Neural Information Processing Systems 36 (2024).

---

### Official Review · Reviewer_Un7H · 2024-11-04

**Soundness:** 3
**Presentation:** 4
**Contribution:** 2
**Rating:** 6
**Confidence:** 4

**Summary:**

This manuscript introduces Interventional State Space Models (iSSM) to reveal latent neural dynamics under causal manipulations, addressing system identification problems in traditional state space models without interventions. The authors gave rigorous proof about the identification conditions and assumptions. They also showed that iSSM can recover true latent dynamics and predict responses to novel perturbations, using both simulated and real biological data from mouse and macaque experiments.

**Strengths:**

- Idea of intervention helps the identification problem is well justified
- Detailed summary of the literature
- Very clear presentation of the derivation process
- Promising results in recovering latent dynamics structures in simulations (i.e. Fig.2)

**Weaknesses:**

- Model assumption 2 is too strong: From the equation on line 191. It looks like if there's a perturbation affects all latent dimensions, the internal dynamics (term A) at that time step would be totally ignored. This is certainly not a conventional way to model linear systems with input. Please give enough justifications on choosing this specific modeling equation.
- This setup is essentially equivalent to system identification in a controlled system where controllability (thus identificabililty) is related to the structure of input matrix B. Could you comment on your assumptions on B? Or was B also inferred during the process?
- Missing technical details in the inference procedure for readers unfamiliar with variational inference and reparameterization through recognition works.
- For inference results, reconstruction correlation seems to be low. Could you include some baseline method correlation? Or how about using the non-intervened SSM for reconstruction.

**Questions:**

- From the derivation in Session 3.3, it's unclear to me how the addition of input terms could help the identification problem. Could you emphasize the non-identifiability where B=0?
- Assumption 3.2 assumes the observation function is piecewise linear but the ones used in simulation are not. Could you justify this?
- Assumption 3.4 seems to be too strong by assuming there're perturbation cases at every latent dimension (and maybe for each time step as well?) Could you comment on how much this assumption applies to actual data? How much violation to this assumption could be tolerated?

---

> ### Author Response · Authors · 2024-11-29
> **Response**
>
> > Strengths
>
> Thank you for the encouraging words.
>
> > Model assumption 2 is too strong
>
> Please check section **4** and **5** of our general response for related discussion to this point.
>
> > This setup is essentially equivalent ...
>
> The control theory literature normally models the interventional inputs as additive. The main novelty of our work is to incorporate interventions in a causal manner. The intuition of why this is helpful is that every time a new intervention is performed, a submatrix of the dynamics matrix senses the data which makes model identification easier compared to classical control theoretical results. We originally had a discussion on connections with control theory in the paper. Since our work connects to a few separate fields we removed that section to make it easier to follow.
>
> > Missing technical details in the inference
>
> Thank you for the suggestion, the variational scheme we’ve presented is quite popular and standard. We’ve referenced a paper that discusses the details of this variational scheme for SSMs in the paper. Still if you think it’s helpful to have the details in the paper we’re happy to include a section in the supplementary material in the camera-ready paper if we get accepted.
>
> > For inference results, reconstruction correlation
>
> This is a good suggestion, please see section **1** of our general response.
>
> > From the derivation in Session 3.3
>
> The addition of input terms mimics a do-intervention. If there is no intervention, i.e. $\boldsymbol{B}\boldsymbol{u_t} = 0$, then each node of the latent dynamical system evolves as usual $\boldsymbol{x_{t+1,j}} = \boldsymbol{A_j} \boldsymbol{x_{t,j}} + \boldsymbol{\epsilon_{t,j}}$. when there is intervention on a node $j$, that is $\boldsymbol{B}\boldsymbol{u_t} \ne 0$, then the j-th node is decoupled from the other nodes and is set to the value $\boldsymbol{B}\boldsymbol{u_t}$, namely $\boldsymbol{x_{t+1,j}} = \boldsymbol{B}\boldsymbol{u_{t,j}} + \boldsymbol{\epsilon_t}$. In the meantime, all other nodes evolve as usual $\boldsymbol{x_{t+1, j'}} = \boldsymbol{A_j} \boldsymbol{x_t} + \boldsymbol{\epsilon_{t,j'}}$.
>
> While we in general do not know which latent node is intervened, this decoupling of the intervened node and the un-intervened nodes under do-interventions offers signatures for identifying each latent node.
>
> When $\boldsymbol{B}\boldsymbol{u_t} = 0$, then all latent nodes evolve following a usual linear dynamical system $\boldsymbol{x_{t+1}} = \boldsymbol{A}\boldsymbol{x_t} + \boldsymbol{\epsilon_t}$. Without interventions, one can find sets of latent variables that can perfectly describe the observed data.  In particular, suppose the data is generated by $\boldsymbol{y_t} \sim P(\boldsymbol{y_t} | f_{\theta}(\boldsymbol{x_t}); \boldsymbol{x_{t+1}} = \boldsymbol{A} \boldsymbol{x_t} + \boldsymbol{\epsilon_t}$ with the true latent nodes $\boldsymbol{x_t}$, then it can also be equally well described by another set of latents $\boldsymbol{\hat{x_t}}$ that is linear transformation of the true latent, i.e. $\boldsymbol{\hat{x_t}} = \boldsymbol{M} \boldsymbol{x_t}$ for all t: $\boldsymbol{y_t} \sim P(\boldsymbol{y_t} | \hat{f_{\theta}}(\boldsymbol{\hat{x_t}})); \boldsymbol{\hat{x_{t+1}}} = \boldsymbol{\hat{A}}\boldsymbol{\hat{x}_t} + \boldsymbol{\hat{\epsilon}_t}$, where $\hat{f} = f \circ \boldsymbol{M}^{-1}, \boldsymbol{\hat{A}} = \boldsymbol{A} \cdot \boldsymbol{M}^{-1}$. Hence the latent $\boldsymbol{x_t}$ is not identifiable under only observational data, i.e. when $\boldsymbol{B}\boldsymbol{u_t}=0$.
>
> > Assumption 3.2 assumes the observation
>
> While Assumption 3.2 assumes piecewise linear, it can be replaced by other assumptions on the mixing function $f$. The key utility of Assumption 3.2 is to achieve linear identifiability of the latent nodes. It can be achieved via piecewise linear mixing function (as in Assumption 3.2) or polynomial function (as in Assumption 4.2 in [1]) or injective differentiable functions (as in Assumption 1 of [2]).
>
> > Assumption 3.4 seems to be too strong
>
> Please see section **4.2** in our general response.
>
>
> **References**
>
> [1] Ahuja, Kartik, et al. "Interventional causal representation learning." International conference on machine learning. PMLR, 2023.
>
> [2] Buchholz, Simon, et al. "Learning linear causal representations from interventions under general nonlinear mixing." Advances in Neural Information Processing Systems 36 (2024).

---

> > ### Comment · Reviewer_Un7H · 2024-11-30
> >
> > **Assumption 3.4 seems to be too strong...**
> >
> > From the response, I still don't see how the sparsity of B could affect the validity of do-interventions assumptions. Could you elaborate more on this point? Even it's hard to justify such assumption, it's also a good piece of work because the algorithms seems work well on real data.
> >
> > Other than that, my concerns are mostly addressed.

---

> ### Author Response · Authors · 2024-11-30
>
> Thank you for the prompt response and apologies if our description wasn’t clear.
>
> First, we make the assumption that each latent causally links to a sparse set of neurons. As we discussed in the general response, this assumption (referred to as sparse coding in neuroscience literature) has neuroscientific roots and therefore is well-motivated. The sparsity assumption translates into the stimulation matrix $\boldsymbol{B}$ being sparse since each column corresponding to a latent has a sparse set of active rows.
>
> Second, when we intervene on individual neurons, only their corresponding latents (which again is a sparse set) are activated. Therefore, interventions on individual neurons translate into do-interventions on subsets of latents. Our theoretical result—identifiability of the latents, dynamics, and emissions using do-interventions—is equivalent to block-identifiability if the interventions activate sparse subsets of neurons. Therefore, if we designate multiple latents per group of functionally coherent neurons, the model identifies those groups of latents. This is still a valuable result since compressing neural dynamics into a smaller set of disentangled latent groups offers interpretability, reproducibility, and the potential of scientific discovery.
>
> Third, given the sparse coding assumption, enforcing the sparsity on matrix $\boldsymbol{B}$ allows for its recovery as well as identification of the latents and model parameters. This is supported by our new results on the consistency of inferred $\boldsymbol{B}$ matrices across random initializations.
>
> That said, as we mentioned in our general response, the model still allows for directly uncovering the causal relationships between neurons by setting the $\boldsymbol{B}$ matrix to identity. In this case having access to interventional datasets where individual neurons are intervened on is equivalent to performing do-interventions on individual latents. We added a new result to the appendix of the paper (Fig. 7) to showcase this application on synthetic models of working memory.

---

### Official Review · Reviewer_gNbJ · 2024-11-04

**Soundness:** 4
**Presentation:** 3
**Contribution:** 3
**Rating:** 6
**Confidence:** 4

**Summary:**

This paper addresses the problem of modeling neural dynamics under causal perturbations, an important open problem in computational neuroscience. The authors propose an extension to PfLDS — an established and expressive state space model for neural data with linear dynamics and nonlinear emissions — that accounts for causal interventions on the latent state of the model, in both the generative model and  inference approach. Furthermore, they provide a proof, under specific assumptions, of identifiability of the unperturbed model dynamics when using causal interventions. The authors apply the proposed approach to simulation of two established hypotheses of neural dynamics underlying motor activity generation, showing they are able to recover the latent trajectories and reconstruct observations under the model. They also apply the approach to two neural datasets involving causal perturbations, showing their approach can be used to gain insight when modeling real neural data.

**Strengths:**

Overall, this is a good paper with a simple and elegant approach to making progress on an important open problem, despite some limitations.

* The paper addresses an extremely important and under-explored open problem in computational neuroscience — modeling neural dynamics under causal perturbation — and as such has potential for large impact in the field. Applying causal perturbations is a critical experimental approach to gaining insight into neural mechanisms that continues to develop with relatively little support on the modeling side. State space models (SSM) are an important and widely used neural modeling approach, and so an important entry point into this problem. Furthermore, the authors attempt to formally address identifiability of the model under causal perturbations, an important open theoretical problem that complicates the practical use of state space modeling in neuroscience.

* Discussion of the scientific background and problem, relevant literature on interventional models and non-identifiability issues is generally excellent.

* The overall approach is clever and well executed. The paper uses a well established, simple but highly expressive SSM model (fLDS), to make progress on this important problem. It suggests a simple and elegant extension to the model to consider causal interventions. There are important and noteworthy limitations of this approach which are not addressed in the paper (see weaknesses), but it is nonetheless an important contribution.

* Inclusion of the interventional structure by reuse of the generative interventional effect parameters B, in the inference process, is a nice idea that aligns with, but stops short of fully structured VAE (eg SVAE) inference approaches.

* The proof of identifiability presented is rigorous and instructive and builds on recent theoretical results across the machine learning field — although the assumptions the proof relies on may be problematic, adding to the unaddressed limitations of this work (see weaknesses).

* Simulations of two dynamical regimes inspired by established motor cortical dynamics hypotheses are well motivated by the literature.

* The paper uses two interesting and relevant neural datasets with causal perturbations, spanning calcium imaging and electrophysiology in mice and monkeys, showing the approach can be applied successfully to data and provide scientific insight.

**Weaknesses:**

While an important contribution, the paper has important weaknesses which should be addressed - and which could justify raising the score for the paper.

**Modeling interventions at the latent level seems to limit interpretability, and raises questions re. assumptions of the identifiability proof.**

A key, unaddressed limitation of the work is the assumption that interventions at the neuron level affect the latent dynamics directly. This is a reasonable approach for making progress on this open problem using SSMs, and it remains an important contribution despite these limitations, but the paper would benefit from explicit consideration of the limitations of this assumption and its implications for the model, the provided proof, and data analysis, which seem to me to be meaningful.

An assumption of the SSM framework is that high dimensional activity can be well described, or summarized, by dynamics in a low dimensional latent space, and the generative model treats observations as emissions from the latent state, ie the causal graph of the model flows from latents to observations and not the other way around. This is a meaningful issue in using SSMs to model experimental causal perturbations performed at the neuron level.

In section 3.1, the authors explain their modeling choice in a slightly confusing / misleading way:
“whenever a neuron is perturbed, its activity is dissociated from all its upstream neurons. This assumption is easy to incorporate in a linear model, which is achieved by ignoring the columns in the dynamics matrix corresponding to the perturbed neuron”

The issue is that the perturbed neuron is not directly represented in the latent dynamics matrix $A$, but rather is “mixed in” by the emission function $f$. In a system with linear emissions, this emission function is directly invertible; in the nonlinear case considered here, it generally is not. In either case of SSM, neurons are not privately represented in specific dimensions of the latent space by design.

The switching mechanism proposed by the authors $1 \lbrace Bu=0 \rbrace$ applies interventions to dimensions of the latent space rather than to individual neurons. Notably, there is no mechanism to encourage the latent dynamics to isolate perturbed neurons in private dimensions of the latent space, and it’s anyway not clear that this would be desirable from the perspective of modeling the spontaneous population dynamics — this is a feature and not a bug of SSMs.

Indeed, the results in eg Figures 3,4 suggest that the learned model does not isolate dimensions of the latents, and instead learns interventional effects across both latent dimensions for every intervention.

From this perspective, the model can really be seen as a fully switching linear dynamics model, with the switch gated by the existence of interventional inputs across all dimensions, and the interventional regime (given by input-driven dynamics Bu) ignoring the current latent state.

Concretely, this choice and its limitations should be made more clear in the presentation of the approach.

From the perspective of modeling usefulness and interpretability --

-There is no way to bake in known elements of the stimulation, eg in the mice data case where specific neurons are targeted based on their response selectivity.
-Can this framework be used to analyze or make claims about the circuit mechanism involved in interventions, ie how individual neurons affect other neurons in the circuit? Can the framework be used to reproduce or further understand the local neuron level effects of stimulation?

**Assumptions of the proof**

Assumption 3.4 in section 3.3: “(do-interventions on each latent node). There is at least one do-intervention (i.e. non-random $u_t$) being performed on each latent dimension of $x_t$.”

In the current wording it’s not immediately clear that these interventions are required to be *separate* ie that the interventions happen on each latent dimension in isolation. My understanding is that this is in fact the case, and that the proof proceeds by considering interventions $1 \lbrace (B u_{t,j} ) = 0 \rbrace = 0$ performed on individual dimensions of $x$, which uncouple $x_{t+1,j}$ from $x_t$, and then arguing that if some dimension $\hat{x_j}$ is also uncoupled by the isolated intervention, than it must be an affine transformation of $x{_j}$.

Does the proof therefore require the ability to intervene separately in individual dimensions of the latent space? That is, if every intervention involves $1 \lbrace Bu_{t,j} = 0 \rbrace = 0 \quad \forall j$, does the proof still work? Presumably this makes it such that all interventions will impact all latent dimensions?

This question relates to the main weakness discussed above - if neuron level interventions are modeled as direct interventions in the latent space that span the latent dimensions and are not isolated, these interventions will not allow isolation of individual latent dimensions in the model, and will be inconsistent with the identifiability proof.

If this is in fact the case, I believe the proof is still instructive and the method is still valuable, but it is a major limitation of the approach when applied to neural data and should be discussed in this light.

**Lack of comparison to non-interventional PfLDS model**

Another important but easy to fix limitation of the paper is the lack of comparisons - how much does the interventional framework improve the ability of the PfLDS model to recover the system parameters, latents, reconstruct observations under novel perturbations etc? An easy win for the paper would be to provide an example in which the interventional data, in both simulation and neural data, is fit using the standard PfLDS model. This would provide a baseline against which to judge the benefits of this approach.

**Showing that generative dynamics are consistent with inferred latents**

Using an LSTM for the recognition network allows for powerful inference, but given that the inference model learns an implicit model of the forward dynamics, it is important to show that generating sequences from the dynamics model {A,B} along with external inputs closely tracks inferred latents.

**Figures could be improved**
* Figures are a bit hard to read and could be dramatically and easily improved, especially figures 1 and 2, 3 (eg axis ticks on panels C,D) and 4 to a lesser extent. Labels have a kind of “bleeding” bold font, text is too small, arrows are fuzzy, indices in the graphical model aren’t clear etc.
* In Figure 1, “spontaneous activity” may be a better label than “resting state” which has other connotations in the neuroscience literature, especially since the activity measured may typically involve the animal acting in an environment, processing experimental stimuli etc, although there is no neural intervention.
* Figures 2D,E are a bit hard to parse with the red black and blue dots and may have some mistakes in the legends? the text says:
(D,E) True (black) and inferred (red) latent (D) and observation (E) dynamics
But the titles are the other way around. Also I believe F represents the “behavior” output of the system, using the term emissions here is confusing since otherwise this term refers to the observations (neurons of the system).

**No way to capture time varying interventional dynamics**

If optical or neural stimulation effects have a time course, this model will be forced to fit the average effect over that time course, unless the modeler decides to design the interventional input U as a time varying signal. This is fine but likely a meaningful limitation that should be noted.

**Questions:**

* See above weaknesses re questions on the interpretability implications of intervening on the latent state, as well as the proof assumptions
* How does the amount of interventional data vs not interventional data differ in effecting reconstruction in simulation? Figure 2G shows for example that reconstruction of both latents and observations improves with the stimulation count, but it’s not clear whether that means the model is fit with more data in this case or not? Would be interesting to show how results are affected by additional observational vs interventional data.
* Observations reconstruction seems to fall for the rotational dynamics case - any insight on why this may be?
* In the simulations, adding interventions improves recovery of latents and reconstructions, but its not clear that the interventions are actually necessary to disambiguate the two possible hypotheses on the motor flow fields? Would be nice to show this.

---

> ### Author Response · Authors · 2024-11-29
> **Response**
>
> > Strengths
>
> Thank you very much for the encouraging words, our thinking is very much aligned with these statements.
>
> > A key, unaddressed limitation of the work is the assumption that ...
>
> This is a great suggestion, please see section **4** and **5** of our general response which might partially address this comment.
>
> > An assumption of the SSM framework is that high dimensional ...
>
> Indeed modeling interventions as affecting latents vs. neurons has been a critical question for us during the development of the model. Please see **3** and **5** in our general response for a more detailed discussion.
>
>
> > The switching mechanism proposed by the authors
>
> Thanks for pointing this out. As we discuss in section **5**  of our general response, in principle the model can be used to capture individual neural interventions through considering one latent per neuron and setting $\boldsymbol{B}$ to identity. But that requires datasets where individual neurons are perturbed in near perfect isolation from other neurons. At least we are not aware of such a dataset yet. We would like readers to think of latents as “groups of neurons” or “micro-circuits” that are functionally similar. In a sense a causal latent variable model such as iSSM can discover such grouping in an identifiable way. Our theoretical results suggest that in the cases where $\boldsymbol{B}$ is unknown but sparse, enforcing sparsity leads to its identification. Inspired by this, we add a sparsity penalty to $\boldsymbol{B}$. We explain this further in section **4** of our general response.
>
> > There is no way to bake in known ...
>
> Section **4.1** of our general response tries to address this comment.
>
> > Can this framework be used to analyze ...
>
> In section **5.2** of our general response we discuss this. We also add a new experiment in the appendix (Fig. 7) to address this comment.
>
> > Assumptions of the proof
>
> Indeed our current proof requires that the interventions are required to be separate for different latents; i.e. the interventions happen on each latent dimension in isolation. If each intervention involves do-interventions on multiple latent nodes, the proof can be extended, under assumptions on the diversity of nodes being intervened in each intervention (e.g. assumption 3 in [1]). We are actively working on these extensions and hoping to present it in future work. Please see also the “entanglement of the latents” section in our generic response.
>
> > Lack of comparison to non-interventional PfLDS model
>
> Thank you for this suggestion, we’ve added comparisons against PfLDS (referred to as SSM) in the revised paper. Please see section **1** of our general response.
>
> > Showing that generative dynamics are consistent with inferred latents
>
> While this is a really interesting suggestion, we think this is an unsolved problem in SSMs that use variational inference schemes. When we condition the model on the observations $\boldsymbol{y_{1:T}}$, $\boldsymbol{u_{1:T}}$, the recognition network provides us with latents at all time points $\boldsymbol{x}_{1:T}$ which are used to generate sequences. When generating sequences from the generative model, we only have access to $\boldsymbol{A},\boldsymbol{B}$ which might not be sufficient to capture all the nonlinearity. We’ve done the experiment you’ve asked for, the generated sequences from the inference network and generative model have similarities, but the gap is still considerable (as is for other SSM models [2,3]). We’ve included this result in Fig. 5 in the appendix.
>
>
> **References**
>
> [1] Ahuja, Kartik, Amin Mansouri, and Yixin Wang. "Multi-domain causal representation learning via weak distributional invariances." International Conference on Artificial Intelligence and Statistics. PMLR, 2024.
>
> [2] Yao, Yuling, et al. "Yes, but did it work?: Evaluating variational inference." International Conference on Machine Learning. PMLR, 2018.
>
> [3] Yao, Yuling, and Justin Domke. "Discriminative calibration: check bayesian computation from simulations and flexible classifier." Advances in Neural Information Processing Systems 36 (2024).

---

> ### Author Response · Authors · 2024-11-29
> **Response (cont'd)**
>
> > Figures could be improved
>
> Thank you for these suggestions, we’ve fixed these in the revised manuscript and we’ll further improve them in the camera-ready paper upon acceptance.
>
> > No way to capture time varying interventional dynamics
>
> **Short-term temporal effects:** If the interventions have temporal effects consistent with the generative model, then theoretically the model should capture it. When the interventional input is omitted, its effect still persists through the value of latents that are affected by it, and will continue propagating in the network through the dynamics and emission models (or the approximate recognition model). Therefore the statement that “the model does not capture time-varying interventional dynamics” is not entirely correct.
>
> **Long-term temporal effects:** The reviewer is indeed correct that if the effect of intervention is long-lasting with complex long-term dynamics (e.g. if interventions cause plasticity effects that change the long-term behavior of the system) the model is not designed to capture it. However, we argue that in that case fitting multiple iSSM models to different chunks of data splitted by the expected time course of learning can be one way to bypass this issue.
>
> That said, we agree that capturing complex interventional temporal effects is still an open question and we hope to study this in more depth in future work.
>
> > How does the amount of interventional data ...
>
> In the experiment in Fig. 2 the total amount of data (i.e. number of trials and the length of each trial) is kept constant across all runs. We change stim_count on the x-axis while keeping the amount of data fixed.
>
>
> > Observations reconstruction seems to fall
>
> The effect is not statistically significant (notice that the error bars represent standard error of the mean). Our conclusion of Fig. 2 G-P is that both observational and interventional data allow for a good reconstruction of data, but only interventional data allows for capturing the true latents.
>
> > In the simulations, adding interventions improves recovery
>
> This is a great suggestion, we’ve done a new experiment to address this. In the experiment we fit iSSM to the interventional data from the two systems, and input the emission models with a grid of latents. If the models faithfully capture the flow fields, the generated trajectories for the given input should be different. This is indeed the case as we present in Fig. 6 in the appendix of the revised manuscript.

---

### Author Response · Authors · 2024-11-29
**Rebuttal**

Thank you for your careful reviews and thoughtful comments. We’re glad that some of you found the paper well-motivated.  As reviewer *gNbJ* mentioned, the problem we study here—modeling neural dynamics under interventions—is quite important but understudied. We believe causal inference is the right tool for exploring this problem, but unfortunately, there has been a disconnect between causal inference and computational neuroscience communities. One of the aims of this paper is to spark interest in the two communities for collaborative work and discussion. That said, we sincerely appreciate your constructive critiques and agree with many of them. Below we try to address as many as we can despite the short rebuttal period. Given that the paper has a mix of theoretical, algorithmic, and neuroscientific contributions we urge the reviewers and AC to take this into account for their final deliberation of the paper.

Common critiques
----------------

**1. Comparison against observational models**

*   **For interventional data:** In the presence of interventions, one could model interventional responses as additive (non-causal) or causal. We refer to the former as SSM and the latter as iSSM (our contribution). We agree with the reviewers that an important missing comparison in the paper is comparing SSM with iSSM on interventional data. To address this, we've done this comparison for all experiments in the paper and modified the figures accordingly.

    *   We extended our code package to incorporate non-causal stimulation effects (here's an [anonymized version of the code package](https://drive.google.com/file/d/1Rqw2wSIdIsRARo6puf0bxGmh5jEX9CVo/view?usp=sharing) for your consideration).

    *   We performed a series of new experiments comparing SSM and iSSM presented in Fig. 2G-P, Fig. 3F, Fig. 4G of the revised paper.

*   **For observational data:** Please note that in the absence of interventions, iSSM and SSM are equivalent. Therefore in Fig. 2 as the stim\_count goes to zero we recover the observational regime. Those comparisons already exist in the previous version of the paper.


**2. Cross-validation results**

*   We’ve performed new experiments on real datasets where we varied the latent dimension and the sparsity hyperparameter for the $\boldsymbol{B}$ matrix and investigated the performance of SSM and iSSM under these changes. These new results are presented in Fig. 3F,H and Fig. 4G,H.

---

> ### Author Response · Authors · 2024-11-29
> **Rebuttal (cont'd)**
>
> **3. Low dimensionality of the dynamics**
>
> *   Indeed as we discuss in the introduction and schematize in Fig. 1, we think of interventions as forcing the neural state out of its low-dimensional attractor manifold. Given this, it is a valid question of how we are still assuming low dimensionality. Our argument is the following: (1) While the observational data is low-dimensional, we think that interventional data is higher dimensional but still lives in a much lower dimension than the number of neurons. This is simply because certain configurations of neural states are not biologically possible \[1\]. (2) In our model, we place a centered Laplace prior on $\boldsymbol{B}$ with the scale hyperparameter $s$ to encourage its sparsity which is critical for identifiability as discussed in the literature \[2,3\]. The optimal latent dimension for our experiments depends on the sparsity of the $\boldsymbol{B}$ matrix as well as the amount of data that we have. With larger datasets and more interventions, our conjecture is that increasing the number of latents always provides better test accuracy.
>
>
> **4. Entanglement of the latents**
>
> *   **4.1 Constant vs. inferred $\boldsymbol{B}$:** Our code package allows for both inferring the stimulation matrix $\boldsymbol{B}$ and setting it to a previously known constant matrix. If we have precise information about the relative location of perturbation sites with respect to neurons, we can encode that information into the matrix $\boldsymbol{B}$ by setting $\boldsymbol{B}_{ij} = D(\text{stim}_i, \text{neuron}_j)$. With this approach, every time a neuron or a group of neurons are perturbed, their corresponding latent is activated which in turn silences or activates downstream neurons in isolation of their other parent neurons.
>
> *   **4.2 The sparsity of $\boldsymbol{B}$:** If we don’t have prior information about $\boldsymbol{B}$, our approach is to add a sparsity penalty by considering a Laplace prior over the elements of $\boldsymbol{B}$. In this case, as we increase the effect of the prior, the latents become more and more disentangled while still fitting the data. We intentionally used a weak prior for the $\boldsymbol{B}$ matrix in our experimental results since we used a small number of latents. With a sparser $\boldsymbol{B}$ in order to fit the data well we need more latents. We added a new experiment where we increased the latent dimensions and used a stronger prior for $\boldsymbol{B}$. The new results are presented in Fig. 3H and Fig. 4H. Our main argument is that if there are multiple iSSM models consistent with interventional data, assuming sparsity of $\boldsymbol{B}$ will result in identifiability. From a neuroscientific perspective, this means that the latents are activated by sparse (and non-overlapping subsets of neurons) which is a sensible assumption for some use cases \[4\].
>
>
> **5. Intervening on latents vs. neurons**
>
> *   **5.1 Neuroscientific motivation:** Please note that many causal experiments in neuroscience (both optical and electrophysiological) still do not perform interventions at the single neuron level. In some cases, the light-gated proteins (e.g. Channelrhodopsin) are expressed in all the neurons of a subtype, and broad-field illumination is used to activate all neurons of that certain subtype. In other experiments, light-gated proteins are expressed in a sparse subset of neurons, and two-photon lasers are used to activate only those neurons. In these cases, due to the spillover effect of lasers, it’s impossible to activate a dense set of neurons with enough spatial precision. Similarly, for electrophysiological techniques such as micro-stimulation, each electrode targets several neurons located in its vicinity as opposed to a single neuron. Due to these (mostly) experimental limitations, we think modeling interventions as affecting latents is a good first step. Therefore, we think given an intermediate latent dimension the model provides a natural causal grouping of neurons into subsets that are behaviorally relevant.
>
> *   **5.2 Modeling causal relations between neurons:** That said, our model still allows for capturing the effect of intervening on individual neurons. In the case where we’re interested in investigating causal connections between neurons, one approach is to consider one latent per neuron. If the experimental setup allows for stimulating individual neurons, then we can set the $\boldsymbol{B}$ matrix to identity. To showcase this specific use case of the model we added a new experiment identifying the connectivity matrix to test between two models of working memory inspired by \[5\]. In this new experiment, we’ve set the matrix $\boldsymbol{B}$ to identity and used it to discover the causal relationships between neurons (Fig. 7).

---

> ### Author Response · Authors · 2024-11-29
> **Rebuttal (cont'd)**
>
> **References**
>
> \[1\] Jazayeri, Mehrdad, and Arash Afraz. "Navigating the neural space in search of the neural code." Neuron 93.5 (2017): 1003-1014.
>
> \[2\] Moran, Gemma E., et al. "Identifiable deep generative models via sparse decoding." arXiv preprint arXiv:2110.10804 (2021).
>
> \[3\] Lachapelle, Sébastien, et al. "Synergies between disentanglement and sparsity: Generalization and identifiability in multi-task learning." International Conference on Machine Learning. PMLR, 2023.
>
> \[4\] Barth, Alison L., and James FA Poulet. "Experimental evidence for sparse firing in the neocortex." Trends in neurosciences 35.6 (2012): 345-355.
>
> \[5\] Qian, William, et al. "Partial observation can induce mechanistic mismatches in data-constrained models of neural dynamics." bioRxiv (2024): 2024-05.

---

### Author Response · Authors · 2024-12-03

Dear Reviewers,

Thank you once again for reviewing our submission and providing valuable feedback. We have addressed your concerns in our rebuttal and kindly ask that you review our clarifications. Please let us know if there are any additional points we can address before the discussion period concludes.

Thank you!

---

### Note · Authors · 2025-02-18

I have read and agree with the venue's withdrawal policy on behalf of myself and my co-authors.

---

### Meta-Review · Area_Chair_j6La · 2024-12-08

**Metareview:**

This work combines the ideas of causal analysis and state-space modeling to learn neural dynamics underlying cellular recordings. The method is based on the use of interventions to perturb the neural population and create the state space displacements needed to better fit causal dynamics, versus the typical statistical fits often used. The authors applied their method to a number of datasets as well. The reviewers thought the approach was a strength of the work, and were generally positive of the direction of the work. However, there were a number of weaknesses noted. While most were more minor, a significant weakness was a fundamental schism between the core assumptions of the causal analysis and the limitations of current technologies for neural perturbations. To break this concern down further, the requirement of being able to perturb functional units---whether single neurons of groups---is essential to the approach. Current technologies can at best blanket perturb a local population or a set of genetically different cells. This capability might not even be granular enough for perturbing latents that involve many neurons, as the functional units might share genetic material (pyramidal cells are a cell type but encode many functionally distinct variables). A deeper dive into the interplay of these assumptions and how they impact what can be learned from this approach seems important and therefore I am not recommending the work be currently accepted.

**Additional Comments On Reviewer Discussion:**

Sadly there were no significant discussions with reviewers.

---

### Decision · Program_Chairs · 2025-01-22

Reject